# Chronological Incongruences between Mitochondrial and Nuclear Phylogenies of *Aedes* Mosquitoes

**DOI:** 10.3390/life11030181

**Published:** 2021-02-25

**Authors:** Nicola Zadra, Annapaola Rizzoli, Omar Rota-Stabelli

**Affiliations:** 1Research and Innovation Centre, Fondazione Edmund Mach, 38010 San Michele all Adige (TN), Italy; nicola.zadra@fmach.it (N.Z.); annapaola.rizzoli@fmach.it (A.R.); 2Department of Cellular, Computational and Integrative Biology—CIBIO, University of Trento, 38123 Povo (TN), Italy; 3Center Agriculture Food Environment—C3A, University of Trento, 38010 San Michele all Adige (TN), Italy

**Keywords:** divergence, mtDNA, Diptera, phylogeny, saturation, rates

## Abstract

One-third of all mosquitoes belong to the Aedini, a tribe comprising common vectors of viral zoonoses such as *Aedes aegypti* and *Aedes albopictus*. To improve our understanding of their evolution, we present an updated multigene estimate of Aedini phylogeny and divergence, focusing on the disentanglement between nuclear and mitochondrial phylogenetic signals. We first show that there are some phylogenetic discrepancies between nuclear and mitochondrial markers which may be caused by wrong taxa assignment in samples collections or by some stochastic effect due to small gene samples. We indeed show that the concatenated dataset is model and framework dependent, indicating a general paucity of signal. Our Bayesian calibrated divergence estimates point toward a mosquito radiation in the mid-Jurassic and an *Aedes* radiation from the mid-Cretaceous on. We observe, however a strong chronological incongruence between mitochondrial and nuclear data, the latter providing divergence times within the Aedini significantly younger than the former. We show that this incongruence is consistent over different datasets and taxon sampling and that may be explained by either peculiar evolutionary event such as different levels of saturation in certain lineages or a past history of hybridization throughout the genus. Overall, our updated picture of Aedini phylogeny, reveal a strong nuclear-mitochondrial incongruence which may be of help in setting the research agenda for future phylogenomic studies of Aedini mosquitoes.

## 1. Introduction

Mosquitoes (Culicidae) are one of the most successful Diptera radiation. They include more than 3600 species classified in two subfamilies and 44 genera and 145 subgenera [1,2,3]. Because they vector a variety of disease, mosquitoes are still the largest indirect cause of mortality among humans than any other group of organisms. Approximately one-third of mosquito species belong to the tribe Aedini, including 1261 species classified in 10 genera [3]. Aedini species are globally distributed and are vectors of many zoonosis of human and animals including filarial nematodes [4] and many arboviruses such as Chikungunya, Dengue, Zika, Yellow Fever, West Nile [5,6,7]. Aedini species include some of the most invasive and medically relevant mosquitoes: *Aedes aegypti* and *Aedes albopictus* [8,9,10,11,12]. *Aedes aegypti* has mainly spread outside its original African range, although it does not seem capable of settling stable populations in continental climates, such as the European one. *Aedes albopictus*, originally from South East Asia, is instead now reported from every continent and has quickly settled in Europe, China, and other temperate zones [7,13]. Genome resources exist for only these two species of *Aedes* [14,15,16], while whole genome data for other invasive *Aedes* is still lacking. These include *Aedes japonicus* and *Aedes koreicus*, which are quickly invading and establishing, respectively, in central Europe [17] and North Italy [18,19] showing competence for the transmission of many arboviruses such as West Nile virus and Zika virus [8,19,20].

Knowledge of the reciprocal affinities of these and other invasive *Aedes* species and the timing of their evolution is important for various reasons. First, a robust phylogeny is essential to polarize key behavioral and ecological traits, as recently shown by Soghigian et al. [21]. In particular, a phylogeny can identify the sister-species of invasive *Aedes* of health concern. The sister-species shares a common ancestor with the species of interest (is the closest related in the phylogenetic tree) and is very useful for correctly polarizing evolutionary novelties, such as new genes in phylogenomics and transcriptomics studies [22,23]. Second, phylogenies may help to define taxonomy and classification. A recent classification [1] has raised the number of genera from 10 to 79; the genera, however, have been later reduced to 10 [3]. Molecular investigations of Aedini relationships can help to clarify these taxonomical issues. Third, dated phylogenies help to characterize the paleo-ecological scenario in which mosquito radiations happen, thus providing evidence with clues about their pre-adaptations as it has been shown, for example, in *Drosophila* [24,25]. Molecular studies have addressed Aedini evolution by studying their phylogeny using both mitochondrial and nuclear markers. While relationships within the Aedini group has been studied in detail using a multimarker approach [21], the origin of the family and their reciprocal affinity with other Culicinae are not well studied, or have not been addressed because datasets were centered only on Aedini [21]. Furthermore, the stability of clades within the Aedini has never been addressed by comparing different statistical frameworks (e.g., maximum likelihood versus Bayesian), or by employing a different model of replacement (e.g., homogeneous versus heterogeneous [26]). 

One key aspect so far neglected in Aedini phylogenetic studies is the direct comparison of the phylogenetic signal from the DNA of the two cellular compartments: nuclear (nDNA) and mitochondrial (mtDNA). It has been shown that the nDNA and mtDNA may carry different phylogenetic signal and produce conflicting phylogenies, in some cases, because of hybridization events affecting mtDNA [27]. MtDNA substitution rate is typically faster compared to nuclear one; this can lead toward homoplasy caused by site saturation, which in turns may affect the topology and may underestimate the correct inference of substitution rates [27].

Little in general is known about how mtDNA and nDNA conflict for what concern estimation of divergence times. In some case, chronological signal can be consistent between nuclear and mitochondrial genes, as in fish [28] and amphibians [29], with discrepancy just in the shallow time part of the tree. In *Drosophila*, the two types of markers recover similar divergences with mtDNA supporting slightly younger estimates than nDNA [24]. In butterflies, the chronological conflict between nDNA and mtDNA is more marked, although it seems to be restricted to a few species experiencing hybridization [30]. In the above cases, the confidence interval of the divergence estimates using the two type of markers largely overlap. Therefore, the conflict is not statistically significant. The molecular clocks of mtDNA and nDNA have never been systematically compared in Aedini.

A systematic comparison of chronological signal in mosquitoes has never been undertaken. An effort to date the Aedini mosquito using nDNA data in a Bayesian framework [31] recovered the origin of Aedini at 157 (Credibility Intervals, CI: 187–124) millions of years ago (MYA) and a Culicidae radiation at 216 (229–192) MYA. A recent effort using a multigene (nDNA + mtDNA) strategy in a maximum likelihood framework [20] recovered an Aedini origin at circa 125 MYA. In the latter, the diversification of *A. albopictus* from its sister species *A. flavopictus* is circa 25 MYA, while *A. albopictus* and *A. aegypti* common ancestor was set at approximately 55 MYA, a time compatible estimate, but quite distant from that based on whole genomes 71 (44–107) MYA [14]. A recent divergence estimate of Culicinae using complete mtDNA [32] recovered the origin of Aedini at 130 (CI: 101–168) MYA and an *A. albopictus-A. aegypti* split at circa 67 (CI: 55–94) MYA. There are, therefore, certain discrepancies in available literature for what concerns the timing of Aedini radiation.

This work aims at providing an updated picture of Aedini phylogeny and divergence, by disentangling the phylogenetic signal in available genetic markers. We used four nuclear and four mitochondrial genes in a Bayesian framework to study the evolutionary history of the Aedini, their relationship with other Culicinae, and the timing of their origin and diversification. Our results revealed previously under looked incongruences between nuclear and mitochondrial data, for what concerns both their rate of evolution and their posterior divergence estimates. This has an important implication for our understanding of Aedini evolution and more generally for the long-lasting issue of incongruences between mitochondrial and nuclear data in inferring species phylogeny and divergences.

## 2. Materials and Methods

### 2.1. Genes and Taxa Selection

In our study, we employed four mitochondrial coded genes: Cytochrome c oxidase I (COI), Cytochrome c oxidase II (COII), NADH dehydrogenase subunit 4 (NAD4) and 16S, and four nuclear-coded genes: Enolase, Arginine Kinase, 18S, and 28S. We choose these genes after various rounds of literature and blast searches because they were the most evenly distributed through the Aedini tribe and the outgroup. Similarly to other recent Aedes phylogenetic studies [21], the current availability of genes in the database did not allow us to sample more genes. The number of annotated genes in Genebank for Aedes and other mosquitoes species is low, in general no more than 5 or 6 markers per species; many species are characterized by many variants of the same marker for example COI. Annotated genome and transcriptome data was present only for the model organisms *Aedes albopictus* and *Aedes aegypti*. We had to exclude from our gene list the genes encoding for white, hunchback, and Carbomoylphosphate synthase (CAD) because poorly sampled within the Culicidae family. We did not employ Internal Transcribed Spacer (ITS) because of poor and ambiguous alignment between Aedini and its outgroups. Since we were interested in studying the origin of Aedini, poor alignment of Aedini ITS sequence with those of the outgroups could have affected the correct inference of their phylogeny. We sampled genes from the same specimen whenever it was possible; in most cases we concatenate genes from different specimens of the same species. This is the common procedure when concatenating genes for inter-specific phylogenetic studies [21,26]. We followed the nomenclature as in [3,33]. For more clarity, in some of our phylogenetic trees, we displayed in brackets the proposed subgenera. Each chosen gene for all the available Aedini taxa was downloaded from GenBank. To reduce missing data and promote a direct comparison between nuclear and mitochondrial data, we selected a species only if at least two genes represented it in each of the two types of markers (nuclear and mitochondrial). Moreover, we excluded species that seemed to be ambiguously labelled. The final dataset finally was represented by 34 evenly phylogenetically distributed Aedini, plus 10 outgroups sampled from other Culicinae, Anopheline, and other Diptera samples (see Appendix A for the species list). The outgroup sequences were essential to root the Aedini phylogeny and to generate nodes for calibrating the molecular clock. For each of the eight markers, we filtered out ambiguous sequence. We used a fast bootstrap RaxML (see details below) to preliminarily check if sequences were clustering within their expected group (e.g., sequences for Aedini to form a monophyletic Aedini group). Sequence clustering to a different group considered unreliable and was excluded from downstream analysis.

### 2.2. Alignments and Phylogenetic Analyses

We aligned each of the eight genes independently. We aligned protein-coding genes using MAFFT through TranslatorX [34], and non-coding genes using MAFFT directly [35]. Finally, the genes were concatenated using FASconCAT [36] and manually edited to detect a few misaligned sites. We generated three aligned datasets: nuclear, mitochondrial, and concatenated. The nuclear dataset (nDNA) is composed of the concatenation of Enolase, Arginine Kinase, 18S, and 28S; it is 3270 nucleotides (nt) in length. The mitochondrial dataset (mtDNA) is composed of the concatenation of COI, COII, NAD4, and 16S; it is 4224 nt in length. The third dataset (concatenated) is the concatenation of nDNA and mtDNA and is 7494 nt in length. To further study Aedini relationships, we generated a fifth dataset based on the original 6298 nt alignment of [21], increasing site occupancy by using Gblocks at default parameters. The final dataset (named Soghigian) was composed of 71 sequences and 3815 nt with 8% of missing data. Although there is some overlap of genes between concatenated and Soghigian datasets, they substantially differ because of the presence of 4 genes (COII, NAD4, and 16S present in concatenated, while ITS is absent from concatenated) and mostly because of a very different taxon sampling. The concatenated dataset contains various outgroup to the Aedini because the aim was to set the origin of Aedini. Phylogenetic analyses were performed mainly at the nucleotide level using both maximum likelihood (ML) and Bayesian statistical frameworks, using, respectively, RAxML [37] and PhyloBayes [38] or BEAST (see below). The RAxML analyses were performed on all datasets using the General Time Reversible (GTR) replacement model plus four discrete rate categories of gamma (G) and employing 100 bootstrap replicates. PhyloBayes analyses were performed using the same model and repeated using the heterogeneous CAT (plus G) replacement model.

### 2.3. Divergence Estimates

BEAST v2.5 was used to reconstruct phylogenies and to estimate divergence times [39]. We use BEAUti to set the analyses using the following prior information to calibrate the clock. We employed a root prior based on the fruit fly-mosquito split using a normal distribution with mean 260 MYA and a 95% prior distribution to be between 296 and 238 MYA, as indicated by [40]. We employed three minimum calibration points for the diversification of Anophelinae, Culicinae, and Culicidae, using, respectively, 34 MYA, 34 MYA, and 99 MYA, according to the three oldest fossils known for each of these groups [41,42]. These calibrations were used for the mtDNA, nDNA, and concatenated datasets. We run BEAST using a different set of (model) priors and choose the most fitting combination of priors using the Harmonic mean, the Akaike Information Criterion (AICm), the stepping stone (SS), and the Path sampling (PS); for the latter, we used the Path sampler package and set the analysis at 50% burn-in with 40 steps of 500,000 chain length. All other chains were run for 100,000,000 generations until Beast log files indicated proper convergences of all posteriors and the likelihood using tracer1.7 [43]. Divergence estimates in PhyloBayes [38] where done using the same calibration priors described above, a CAT plus Gamma replacement model and a LogNormal relaxed clock.

## 3. Results

### 3.1. Conflicts between Nuclear and Mitochondrial Phylogenies

Bayesian inference of the eight concatenated genes dataset under a homogeneous replacement model (GTR, Figure 1A) reveals a generally well-supported tree with the *Aedes* genus divided into two distinct clades as in [21]: Clade A (in pink, Posterior Probability (PP): 1.00) comprises various species including *A. albopictus* and *A. aegypti*; Clade B (in orange, PP: 1.00) comprises various species often regarded as *Ochlerotatus* plus others referred to as *Aedes* such as *A. koreicus*. Species of the *Psorophora* genus are the sister of Clade A + Clade B (PP: 1.00). Within Clade A we observed two groups (dark pink), one consisting of species attributed to *Stegomya* + *Armigeres* (clade A1, PP: 1:00), the second containing four genera (*Aedimorphus*, *Catageiomyia*, *Diceromyia*, and *Scutomyia*; PP: 0.92). The mutual relationship of no-Aedini Culicinae is instead unresolved (PP: 0.48); four genera (*Sabethes*, *Wyeomyia*, *Malaya*, and *Toxorhynchites*) form however a robustly supported (PP: 1.00) group, which we have provisionally named Group C.

Our concatenated analysis of Figure 1A recovers, at least for most nodes, a robust topology. One of the aims of our study was, however, to disentangle the phylogenetic signal for the Aedini by exploring its consistency over different data types and methodological treatments. We, therefore, analyzed the nDNA and mtDNA datasets separately (respectively, Figure 1B,C), and reveal various instances of mitochondrial-nuclear incongruence. Overall, both trees are less resolved than the concatenated tree (for example, they both do not support Group C nor Group A2), pointing toward the utility of concatenating genes. The nuclear tree is, however, markedly more resolved (it has overall higher supports at nodes) than the mitochondrial one. It does support, for example, the monophyly of *Aedes* and both Groups A and B (all with PP > 0.9), while mtDNA dataset does not support them. These differences may be explained by less phylogenetic signal in the mtDNA dataset. This is, however, not related to fewer nucleotide positions as the mtDNA alignment is larger than the nDNA one (4224 nt vs. 3270 nt). We identify some interesting cases of well-supported incongruences between the nDNA and the mtDNA trees involving *A. albolineatus*, *A. subalbopictus*, and *Toxorhynchites* sp (depicted by arrows in Figure 1B,C). There are various topological incongruences, for example for the position of *A. subalbopictus*, the two *Psorophora* and *Uranotaenia lowii*, but their affinities did not receive high PP in at least one of the two trees, therefore they are not considered statistically significant.

### 3.2. A Conservative Picture of Aedini and Other Culicinae Phylogeny

To explore in more detail the phylogenetic signal behind our Bayesian trees of Figure 1, we further performed phylogenetic analyses employing different statistical frameworks, different model of replacement, and type of datasets (Figure 2). In panel A we depict the result of a Maximum Likelihood (ML) analysis of the concatenated dataset. In panel B is the same dataset analyzed in a Bayesian framework using an among-site heterogeneous CAT model more suitable for ancient radiations and saturated datasets [44].

In panel C is the ML analysis of a dataset (named Soghigian) centered on Aedini and derived from [21]. To provide a conservative picture of Aedini phylogeny, we have collapsed a node if its bootstrap support (BS) from the Maximum Likelihood (ML) analysis was lower than 75% and if its posterior probability (PP) from Bayesian analysis was lower than 0.9. We found a consistent signal (compare Figure 1A with Figure 2A,B) for a group of *Sabethes*, *Wyeomyia*, and *Malaya* (Sabethini tribe), plus *Toxorhynchites* (Toxorhynchitini tribe) which we have provisionally named Group C. This group is monophyletic using both homogeneous and heterogeneous models of evolution, but its internal relationships, as well as its relative affinity with other Aedini, is inconsistent over different analyses and in general not significantly supported. This group is not consistent with a previous multigene phylogeny, which supports *Toxorhynchites* as closely related to *Mymoyia* than to the Sabethini [31]. Although highly supported in all our concatenated analyses, we advocate caution in considering the validity of Group C, as our analyses may have been biased by an unfortunate combination of reduced taxon and gene sampling; indeed, in most analyses, *Toxorhynchites* is the sister taxa of *Mymoyia*, therefore disrupting the monophyly of Sabethini. Our investigations are instead congruent with previous studies [20] in supporting Group A, and to a lesser extent Group A1 (*Stegomya* + *Armigeres*). Group B is instead supported only by the homogeneous GTR model of evolution. Site heterogeneous models of replacement such as CAT have been repeatedly shown as being capable of reducing systematic errors [45]; we cannot, therefore, exclude that the signal responsible for Group B is artefactual and we advocate care in considering it as monophyletic. From a systematic point of view, our phylogenies support the classical 10 genera classification of Aedini [2,20]. Overall, while some nodes are robustly supported in all analyses of Figure 2 (for example, Group A), other nodes are poorly supported or are supported only in one analysis.

### 3.3. Divergence Estimates of the Aedini

To define which evolutionary models better describe the radiation of mosquito in our concatenated dataset, we contrasted the strict clock versus the log-normal relaxed clock models, the coalescent versus the speciation model, and the HKY versus the GTR replacement model (Table 1). A relaxed clock is favoured over the strict clock. Furthermore, under the relaxed clock, the coefficient of variation rate was approximately 0.4 for the mitochondrial data and roughly 1 for the nuclear, further indicating that an uncorrelated clock hypothesis suits better our datasets than a strict clock. This is because if a log-normal clock has a coefficient of variation close to 0, it could be considered clock-like, so comparable with a strict clock [46]. Demographic speciation models are more supported than Coalescent model, but the stepping stone and path sampling could not discriminate between a Yule and a Birth Death model; we chose the Yule model because it was favoured by the AICm, which penalizes based on the number of free parameters. The two models provided nevertheless with similar results (Table 1). Therefore, we used a combination of GTR+G, relaxed log-normal, and Yule models for our clock analyses.

Our analysis of the concatenated dataset using the most fitting models, allows us to obtain a picture of Aedini evolutionary history, which we have contrasted with the appearance of some major vertebrate lineages and flowering plants in Figure 3. According to our posterior estimates, the mosquito family (Culicidae) diversified in its two subfamilies—Culicinae and Anophelinae—approximately 180 MYA (95% High Posterior Densities, HPD 137–228 MYA) in the lower Jurassic. The earliest fossil of a Chaoboridae, the Culicidae sister group, is 187 MYA [42]. This would suggest a very rapid diversification of Culicomorpha. Our estimates tend to match the proposed origin of angiosperm [47]; however their evolutionary history is not clear yet, and the origin of angiosperm could be older than expected [48]. Culicinae diversified in two clades (Culicini and the clade leading to Aedini) between the end of the Jurassic and the early Cretaceous, at 146 (108–182) MYA, while the Aedini tribe diversify at 113 (83–143) MYA with the split of *Aedes* from *Psorophora* genus. Within Aedini, Clade A, and Clade B originated circa 106 (77–133) MYA. Within Clade A, the subgenus *Stegomya* (which includes model organisms *A. albopictus* and *A. aegypti*) originated 84 (58–109) MYA, concomitantly with the diversification of Clade B (which include the subgenus *Ochlerotatus*) at 86 (61–111) MYA in the late Cretaceous. To test for the effect of outgroup on our dated phylogenies, we repeated the analysis of our concatenated alignment, excluding Brachycera outgroup. This additional analysis shows that the calibration point drives our divergence estimates at the root. The median height is younger without outgroups, although the two analyses are compatible for what concerns their (overlapping) 95% HPD (Table 2). The rooted tree provided more precise estimates. The 95% HPD is smaller in the root-calibrated phylogeny then in the unrooted one. Overall, our date estimates tend to be slightly younger than the ones provided previously [14,21,31] for what concern the origin of Culicinae, but slightly older for what concern the origin of Aedini (see Table 2).

### 3.4. Chronological Incongruences between Nuclear and Mitochondrial Data

Clock analysis using separately nuclear and mitochondrial genes (Figure 4) revealed unexpected strong incongruences. The estimates for the origin of the main mosquito clades (deep nodes of the phylogeny) are similar using the two datasets and reinforce our findings using the concatenated data of Figure 3. For example, Culicinae originated in the early Jurassic and Aedini in the Cretaceous both in the mtDNA and nDNA. However, there is a strong discrepancy for what concerns the diversifications within the Aedini lineages. For example, Group A diversified during the Cretaceous using mitochondrial (and also concatenated) data, but is much younger (Paleogene) using nuclear data (Table 2 for details). Even more discrepant is the origin of *Aedes* species: *A. aegypti* and *A. albopictus* split ranges from 81 (61–101) MYA using mitochondrial data, to 30 (15–45) MYA, using nuclear data; the split between *A. albopictus* and *A. flavopictus* is 32 (20–47) MYA using mitochondrial data and just 4 (0.5–11) MYA using nuclear data. Worryingly, those estimates do not overlap at their confidence interval. From a statistical point of view, this indicates that the two datasets reject each other. Overall, the estimates from the concatenated dataset are more similar to those of the mtDNA dataset then the nDNA dataset (Figure 4C–E).

We further inspected the posterior rates of both the nDNA and the mtDNA trees (Figure 5A,B, respectively; tree topologies are similar to those of Figure 1B,C). Additionally, in the case of rates there are various discrepancies between the two datasets: *A. furcifer* and *A. taylori* are, for example, fast-evolving according to mtDNA, but slow evolving using nDNA. These high rates are likely responsible for the dubious position of these two species in the mtDNA tree of Figure 5B. Our clock analyses returned mean posterior evolutionary rate (calculated over the whole tree) of 1.01 × 10^−3^ (sd = 1.12 × 10^−4^) mutation per site per millions of years (msm) for the mtDNA and of 9.93 × 10^−4^ msm (sd = 1.8 × 10^−4^) for the nDNA.

### 3.5. Mitochondrial-Nuclear Chronological Incongruences Are Consistent over Different Analytical Condition

We tested the robustness of the chronological incongruence observed between mitochondrial and nuclear data (Figure 4) by verifying if the results are biased by the taxon sampling and the number of gaps in our alignments. We first repeated the clock analyses using a reduced version of our dataset. We excluded three species (*Chagasia*, *Uranotenia*, *Aedes albolineatus*) which had an extremely different branching position in the phylogeny of the nDNA and mtDNA analysis. Results are very similar compared to when using the full dataset for what concerns both the divergence estimates (Appendix A) and the average mutation rate at branches (Appendix A). This indicates that the chronological discrepancy is not due to the presence of rough taxa in the dataset. We then tested if the pattern we observe is due to a particular taxon and site sampling by repeating the analyses using a different dataset. We inferred divergence estimates using separately the nuclear and mitochondrial partitions of the Soghigian alignment, derived from [21] and previously used for Figure 1C. This dataset is characterized by a different taxon representation compared to our dataset (it is centred on *Aedes* and contains few outgroups) and by a higher site representation (contains a lower amount of missing data, see methods for details). Results (Appendix A) provide a similar picture to when using our nDNA and mtDNA. Divergences closer to the root are similar, but those within *Aedes*, including the diversification of Clade A and Clade B are very different. This indicates that the chronological discrepancy is not due to peculiar taxon or gene sampling nor is affected by the amount of missing data in the datasets.

Because of its higher mutation rate, MtDNA is, in general, more prone to saturation than the nuclear genome [49]. Accordingly, we would expect to underestimate the number of observed mutations in mtDNA dataset compared to the nDNA one with the consequences that nodes using mtDNA dataset should appear younger than they are. We observe, however, exactly the opposite. Saturation and heterogeneity of the replacement pattern may have nevertheless played a certain role in overestimating the mitochondrial age in our mitochondrial phylogeny. We therefore tested our datasets for saturation by inferring divergences under the CAT model, a mixture model known to be less sensitive to systematic error in the presence of site-specific saturations [45]. The CAT trees are indeed slightly different than those obtained using homogeneous models of replacement (Appendix A). The divergences become more similar between the two datasets, but the nDNA dataset consistently return younger age for recent nodes compared to the mtDNA dataset. We conclude that site heterogeneity is only partially responsible for the mitochondrial–nuclear chronological discrepancy.

## 4. Discussion

Our phylogeny of Aedini using a concatenated dataset of eight mtDNA and nDNA markers (Figure 1A) recovers, at least for most nodes, a robustly supported tree topology. Our comparison of mtDNA and nDNA datasets revealed however some unexpected highly supported phylogenetic discrepancies (Figure 1B,C). We suggest three explanations for these incongruences. The first is wrong taxonomic assignment during field collection. Accordingly, one or more genes for some species may come from another (similar and mistaken for) species creating conflicting phylogenetic signal and wrong tree topology. Another, in our opinion less likely, explanation involves complex evolutionary events, such as past hybridization between species, which have resulted in different inheritance patterns for either the mtDNA or some regions of the nDNA. The final explanation is the stochasticity embedded in small (four genes) datasets, such as the ones we have used. The stochasticity in the mtDNA tree may have been exacerbated by systematic errors related to the fast-evolving nature of the mtDNA [49,50] and to evident high level of apomorphies, as revealed by the longer terminal branches in the mtDNA tree compared with the nDNA one. The different phylogenetic signal, however, does not seem to relate to the amount of missing data as the mitochondrial alignment is more complete than the nuclear one (38% of missing data in mtDNA vs. 45% in nDNA). Whatever the source of the topological discrepancies between datasets, our result point toward the limitation of a PCR-scaled approach for Aedini phylogeny and point toward future studies based on whole mtDNA and genome-scaled nDNA dataset. Indeed, undetected stochastic and systematic type of errors may also affect the concatenated dataset, as we have shown that the phylogenetic signal is unstable at many nodes when employing different replacement models and statistical frameworks (Figure 2B and Appendix A). In particular, the poor support using heterogeneous models may be due to the ability of this model to detect saturated or fast-evolving sites [45]. Under this scenario, the highly supported Clade B when using homogeneous GTR model (Figure 1A and Figure 2A) may be the result of a systematic error. The various phylogenetic incongruences we observe using different replacement models (Figure 2A,B) reinforce what we have found when comparing nuclear and mitochondrial data. They alert us of possible systematic and stochastic errors. We advocate to adopt a cautious, conservative way in interpreting our (but also other available [21,31]) trees of the Aedini based on few genetic makers, as seemingly high supports (as in our Figure 1A) are not consistent over data type (Figure 1B,C and Figure 2C), method of inference (Figure 2A) or replacement model (Figure 2B). In perspective, our data indicate that the phylogeny of Aedini should be resolved with confidence only using a genome-scaled nuclear and a complete mtDNA dataset as done in other dipteran studies [24].

Our divergence estimates using both the concatenated and the mtDNA and nDNA datasets (Figure 3 and Figure 4) are concordant in indicating that mosquito radiated from the mid-Jurassic on and that Aedini radiation started in the mid-Cretaceous, quite concomitant with the origin and the earliest diversifications of mammals first, and later birds during the Cretaceous. We cautiously speculate that there may have been a general history of co-radiation (the available data do not provide enough evidence to advocate co-evolution) between the Aedini and warm-blooded vertebrates. In support of this hypothesis, the Aedini group has a specific preference for mammals and birds [51]. The fact that a relaxed clock better fits our Aedini concatenated dataset is not surprising considering that a large variety of ecologically characterizes mosquitoes and demographic habits [21], which can be responsible for different generation times and therefore different branch rates [24]. The mean posterior rate for the mtDNA dataset is 1.01 × 10^−3^ msm, higher than the 9.93 × 10^−4^ msm estimated for the nuclear genes. The higher mutation rate of mitochondrial genes is expected as the mtDNA is well known to evolve faster than the nuclear genome in animals [52]. Our mean mtDNA rate estimates are, however, circa one order of magnitude smaller than the mitochondrial COI rate of coleopterans (1.17 × 10^−2^ msm) inferred by Papadopoulou et al. [53]. This can be explained by different timespan between the latter and our dataset. Indeed, shallow phylogenetic studies consistently provide with faster evolutionary rate than deep phylogenies [54,55]. Our nuclear rate estimates are in line instead with those inferred over long phylogenetic distances using Ecdysozoa nuclear data (mean 1.01 × 10^−3^ msm, [56], but lower than those based on mitochondrial *Drosophila* data (7.9 × 10^−3^) [24]; this indicates that mosquitoes may have been characterized during their radiation by an overall smaller number of generations per year compared to *Drosophila*. We found that the nDNA data of most lineages within Clade A evolves faster than in the lineages of Clade B; this patter is less marked, but conserved in the mtDNA data (Figure 5). A possible explanation for this pattern is that species of Clade A have in general more generation per year than those of Clade B. The two important invasive *Aedes* species, *A. albopictus*, and *A. koreicus* are characterized by markedly higher replacement rate if compared with their respective sister species, *A. flavopictus* and *A. japonicus*; this pattern can be observed for both mitochondrial and nuclear data. Assuming that the instantaneous mutation rate is conserved within the genus, this result suggests that *A. albopictus* and *A. koreicus* are characterized by a higher number of generations per years compared to other closely related *Aedes*, a hypothesis which may at least partially explain their high invasive potential.

Our analyses revealed a consistent chronological incongruence between the phylogenetic signal of nuclear and mitochondrial genes. mtDNA provides divergence times within Aedini significantly older than nDNA. Previous clock studies in insects have shown poor [24] to moderate [30,57] incongruence between nuclear and mitochondrial data. In these analyses mitochondrial and nuclear estimates, although different, were overlapping for what concerns their 95% HPD. In our phylogenies, the 95% HPD do not overlap, indicating a statistically significant incongruence. We have shown that these incongruences do not depend on rough taxa (compare Figure 4 and Appendix A), nor on-site occupancy and gene sampling (compare with Appendix A), although there is a mitigation of the discrepancies when using a heterogeneous model of replacement (compare with Appendix A). We conclude that the mtDNA–nDNA chronological incongruence in Aedini data does not depend on analytical conditions, although the correct interpretation of saturation in both datasets, particularly the mtDNA one may play a certain role. Based on our results, we cannot exclude that there may have been a long history of multiple hybridization events within *Aedes* species which have affected the mitochondrial genome differently than the nuclear one. Indeed, complex phylogenetic signal due to multiple hybridization events has been recently shown in the *Anopheles* mosquitos [58,59]. The observed discrepancies prevent from drawing a conclusion on the actual timing of diversification of model organisms such as *A. albopictus* whose mean split from sister species *A. flavopictus* may dramatically range from 32 MYA using mitochondrial to just 4 MYA using nuclear data. On the light of these results, we advocate that future research should concentrate on determining the biological (or methodological) reason of this discrepancy by comparing timetrees from whole mtDNA genomes with those from genome-scaled sampling of nuclear genes.

In conclusion, we have provided here a detailed analysis of the phylogenetic and chronological signal in currently available nuclear and mitochondrial genes of the Aedini. Overall, our data point toward the limitation of a multigene PCR-scaled approach for Aedini phylogeny and indicate that future research should be based on genome scaled data. Probably our most interesting finding is the strong chronological incongruence between the nuclear and the mitochondrial data. We could exclude various possible misleading factors such as taxa assignment, missing data, and saturation (Appendix A), but could not ultimately test a stochastic effect related to using only eight genes. This is because at present there is not enough data in databases to build a taxon-rich genome-scaled dataset centred on Aedini. The incongruences we have identified do not currently allow defining the exact timing of evolution of important model organisms, such as *A. aegypti* and *A. albopictus* [60]. We advocate that these chronological incongruences should be investigated in future by comparing whole mitogenomes with genome-scaled nuclear data as we have done for example, in *Drosophila* [30].

## Figures and Tables

**Figure 1 life-11-00181-f001:**
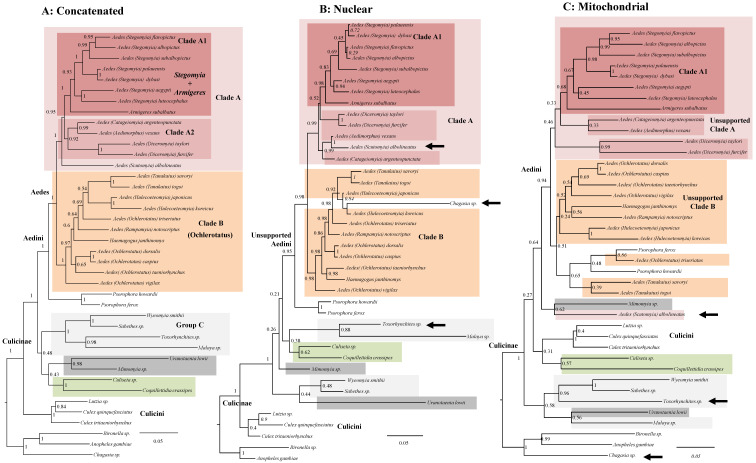
Topological incongruence between mitochondrial and nuclear data. (**A**): Bayesian consensus tree of concatenated dataset. (**B**): Bayesian consensus tree of nuclear (nDNA) dataset. (**C**): Bayesian consensus tree of mitochondrial (mtDNA) dataset. All analyses have been performed using a GTR+G model in PhyloBayes. Numbers at nodes are posterior probabilities (PP). Groups identified using the concatenated dataset have been colored. Arrows indicate highly supported incongruences between the nDNA and mtDNA datasets. The corresponding Maximum Likelihood trees are in Appendix A.

**Figure 2 life-11-00181-f002:**
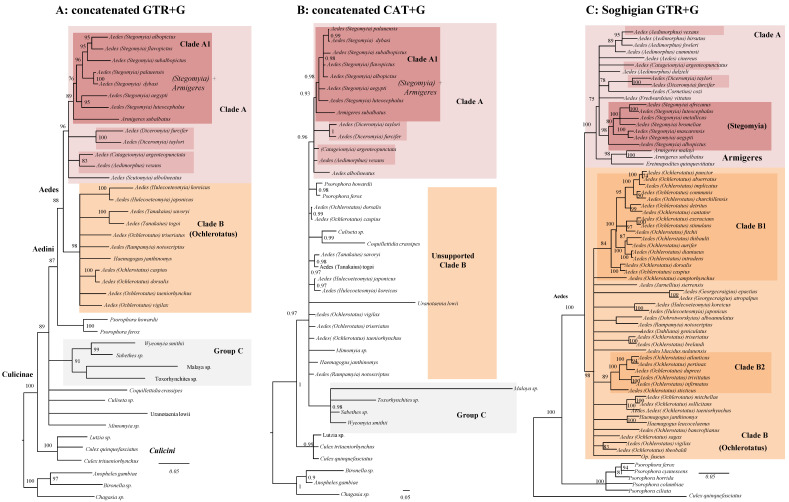
A conservative picture of Culicinae phylogeny using different models and datasets. To highlight lack of phylogenetic signal, all nodes below PP 0.90 and BS 75 have been collapsed. (**A**): Maximum likelihood tree of the concatenated dataset using the GTR+G model in RaXml. (**B**): Bayesian consensus tree of the concatenated dataset under the CAT+G model using PhyloBayes. (**C**): Maximum likelihood tree of a modified [21] dataset under the GTR+G model using RaXml. The number at nodes are bootstrap supports (BS) in panels A and C, and posterior probabilities (PP) in panel B. The backbone of the tree and the monophyly of Clade B (orange) are strongly supported using GTR, but not using CAT. Many relationships within Clade B are poorly supported in all analyses. Full trees with all nods and supports are in Appendix A.

**Figure 3 life-11-00181-f003:**
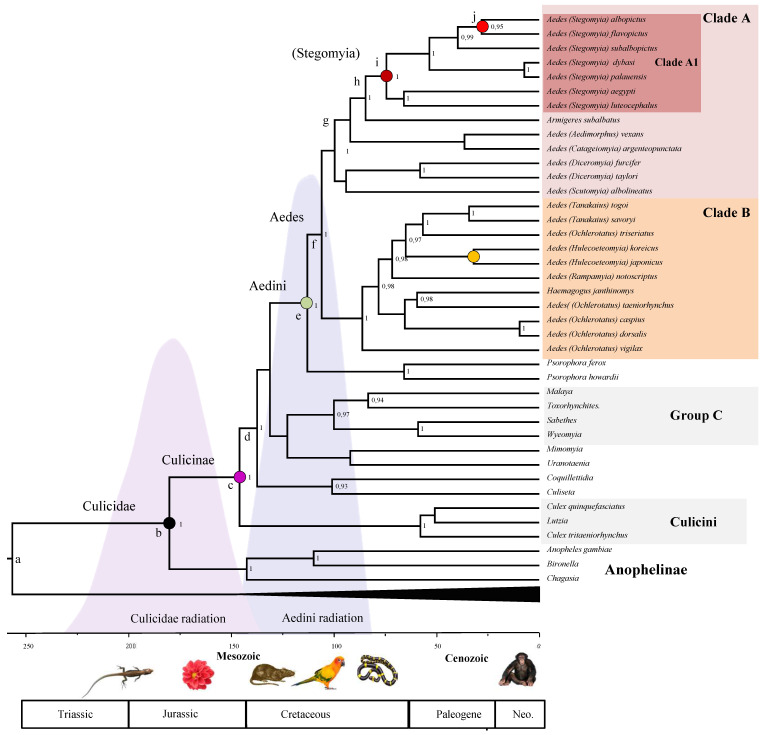
A Bayesian estimates of Aedini divergence. Posterior consensus tree from the analysis of the concatenated dataset. The two shaded distributions highlight the distribution of the 95% HPD for the origin of the Culicinae (b node) and the split of the Aedini (e node): for precise estimates and the 95% HPD see Table 2. Supports at nodes are posterior probabilities higher than 0.95. Time is in millions of years before the present.

**Figure 4 life-11-00181-f004:**
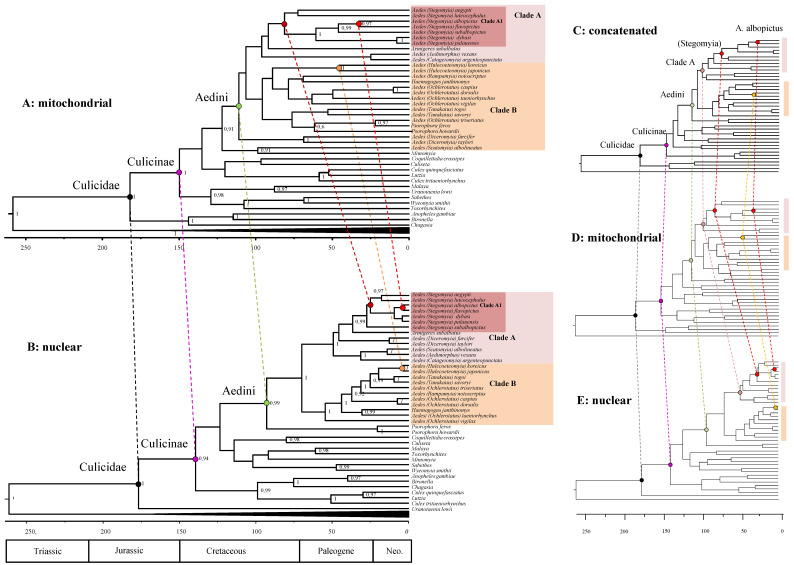
Chronological incongruence between mitochondrial and nuclear data. Note that while posterior estimates are similar for ancient nodes, there are strong incongruences for recent nodes. (**A**): posterior consensus tree from the analysis of the mtDNA dataset. (**B**): posterior consensus tree from the analysis of the nDNA dataset. (**C**–**E**): The concatenated, the mitochondrial, and the nuclear trees simplified for comparison. Supports at nodes are posterior probabilities higher than 0.95. Time is in millions of years before the present.

**Figure 5 life-11-00181-f005:**
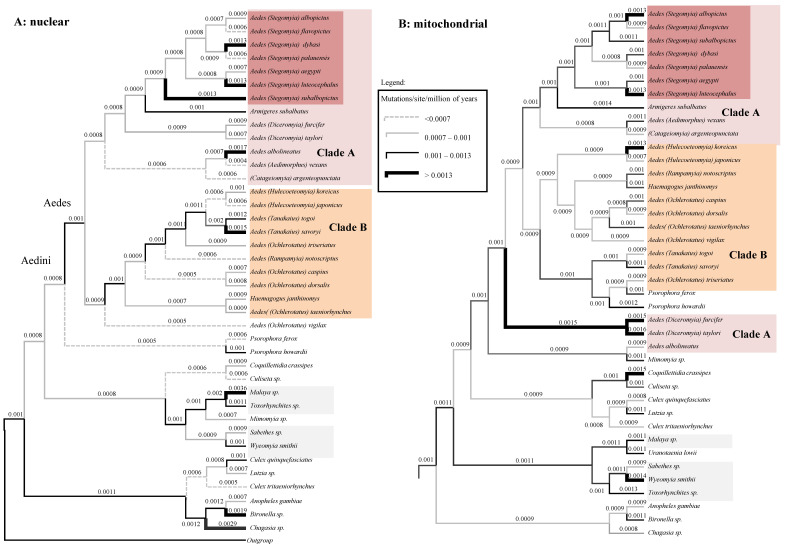
High degree of rate heterogeneity between nuclear and mitochondrial data. (**A**): the nDNA Bayesian phylogeny with mean posterior rates plotted on branches. (**B**): the Bayesian mtDNA phylogeny with mean posterior rates plotted on branches. Note that there are local accelerations of rate (bold lines) in certain taxa only in one of the two data types.

**Table 1 life-11-00181-t001:** Model tested with divergence estimates for two nodes.

Clock Model	Substitution Model	Tree Prior	logLikelihood	AICm	Harmonic Mean	PS/SS	Culicidae	Aedini
Strict	GTR	Yule	54,908.6	109,892.3	−54,926.7	4	156 (114–204)	90 (75–104)
Relaxed (LogN)	HKY	Yule	54,624.8	109,467.9	−54,666.3	5	166 (119–215)	96 (68–125)
	GTR	Yule	54,362.7	109,108.1	−54,424.3	1	180 (137–228)	113 (83–143)
		Birth Death	54,363.6	109,115	−54,414.9	1	180 (135–227)	112 (82–142)
		Coalescent Constant	54,370.2	109,208.7	−54,415.7	3	173 (123–225)	100 (66–132)

**Table 2 life-11-00181-t002:** Divergence estimates of selected nodes from Figure 3 and other analyses. For each node, we provide the mean and the 95% high posterior density. On the right column, we provide estimates from previous studies.

Node	Taxonomic Level	Concatenated Dataset Figure 3	No Outgroup (Concatenated)	Nuclear Data Figure 4A	Mitochondrial Data Figure 4B	Others: Reidenbach09; Soghigian17 *; Da Silva 20 #; Chen 15 ^
**a**	Diptera	257 (223–294)		261 (225–296)	258 (224–293)	260 (239–295) ^
**b**	Culicidae split (Culicinae origin)	180 (137–228)	100 (50–185)	178 (113–245)	182 (143–223)	216 (229–192) 182 # 218 (181–260) ^
**c**	Culicinae split	146 (108–182)	92 (41–139)	139 (92–194)	150(118–184)	204 (226–172) 130 # 179 (148–217) ^
**d**		137 (103–173)	86 (38–127)	123 (79–171)	135 (104–164)	
**e**	Aedini split (Aedes origin)	113 (83–143)	64 (34–122)	92 (55–137)	111 (95–150)	123 (155–90) 125 * 102 #
**f**	*Aedes* split (Clades A-B split)	105 (77–133)	57 (28–110)	69 (42–103)	107 (85–133)	92 (123–61) 102 *
**g**	Clade A split	99 (72–126)	51 (24–100)	49 (29–76)	96 (73–118)	
**h**		83 (59–109)	50 (22–93)	36 (20–57)	92 (71–116)	
**i**	*Stegomya* (*A. aegypti–A. albopictus*) split	73 (50–96)	36 (14–70)	27 (15–45)	81 (61–102)	55 * 67 # 71 (44–107) ^
**j**	*A. albopictus–A. flavopictus* split	28 (14–43)	36 (14–70)	3.7 (0.1–11.2)	33 (20–46)	25 *
**l**	*A. koreicus–**A. japonicas* split	32 (15–51)	14 (3–31)	3.6 (0.2–10.9)	46 (24–71)	20 *

## Data Availability

All data generated for this study are included in this article and its Appendix A.

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
