# Peer review of "Chronological Incongruences between Mitochondrial and Nuclear Phylogenies of *Aedes* Mosquitoes"

_life, 2021, doi:10.3390/life11030181_

Round 1
Reviewer 1 Report
Review of the article titled “Chronological incongruences between mitochondrial and 2 nuclear phylogenies of Aedes mosquitoes” (ID: life-1094817). Zadra et al. present a study on the problems with resolving of phylogeny in Aedes mosquitoes when using mitochondrial and nuclear markers. The idea of this study is not novel as the topic was examined in several previous studies with the most similar of Soghigian et al. 2017. The current study extend previous knowledge on Aedes phylogeny on more molecular markers used simultaneously and more taxa included in phylogenies. This could be an interesting contribution to the field, however, some decisions made by Authors in data collection makes this study speculative and repetitive. Considering my major doubts (listed below), I can not recommend publication of this manuscript.
Specific comments:
- 73 and 439-440
It is too general a statement. There are many parts of nuclear DNA, which are known to mutate faster than mtDNA e.g. microsatellites. Some introns or spacers show a similar rate of mutation.
DNA is not evolving – taxa evolve.
- 113-114
I wonder if these markers (genes) were available from the same samples (specimens) for all or most of genes? If not - how Authors deal with this problem?
Why Authors had not attempted to sequence missing genes for all or majority of Aedes species?
- 125-126
How many samples (specimens) were available for each of examined species?
Based on the look on trees I suppose that only 1 / species. I think that it is not enough for many reasons. Especially if particular sequences were from various samples.
- 128
How that ambiguous samples were determined?
- 325-327
I am surprised that divergence of taxa within the same genus could be so ancient. It is a common phenomenon in mosquitoes or Dipterans? Such ancient dates are determined by calibrations in very distant splits (e.g. flies and mosquitoes) – I am not convinced that such old dates are appropriate for calibration in studies within the single genus.
I suppose that these estimates are not so reliable, especially as mutation rare in mosquitoes had to be c. one order of magnitude smaller than in other insects (see l. 441-442).
Finally, I do not understand information about the divergence of taxa which are not closely related like species: A. aegypti and A. albopictus – these two species had no common recent ancestor, therefore the presentation of split dates for them is strange.
- 395
I think that the whole paper should not be prepared without solving this problem. Currently, all possible explanations are speculative due to uncertainty in correct taxa assignment to sequences.
- 407
I am surprised by so large missing data in the analyzed alignments – this could strongly affect results.
- 423-426, 481-483, 4870488, 492-494
I am surprised that Authors inform so many times that this study should be based on analyses of whole mitogenome or genome scans, what is true. So if they are aware of that need, why this study was not planned and executed on such data instead of the use of uncertain and leaky sequence datasets from previous studies?
In several places are genus and species names not italicized.
Author Response
Review of the article titled “Chronological incongruences between mitochondrial and 2 nuclear phylogenies of Aedes mosquitoes” (ID: life-1094817). Zadra et al. present a study on the problems with resolving of phylogeny in Aedes mosquitoes when using mitochondrial and nuclear markers. The idea of this study is not novel as the topic was examined in several previous studies with the most similar of Soghigian et al. 2017. The current study extend previous knowledge on Aedes phylogeny on more molecular markers used simultaneously and more taxa included in phylogenies. This could be an interesting contribution to the field, however, some decisions made by Authors in data collection makes this study speculative and repetitive. Considering my major doubts (listed below), I can not recommend publication of this manuscript.
>First of all, thanks for your time in reviewing our manuscript.
>We disagree with referee that the manuscript (MS) is “speculative and repetitive”. Previous studies analysed mitochondrial and nuclear markers in concatenation or analysed only one type of makers. None of them, to our knowledge, tried to disentangle the different signal present in the two cytoplasmic counterparts. This issue is well introduced in page 2 “One key aspect so far neglected in Aedini phylogenetic studies is the direct comparison of the phylogenetic signal from the DNA of the two cellular compartments: nuclear (nDNA) and mitochondrial (mtDNA)….”.
>Our study is the first to reveal an unexpected discrepancy between the two types of markers. There is no speculation here: the two markers provide different divergences. We have conducted various analyses to understand the reason behind this discrepancy and show that it is robust to analytical conditions. This is a key finding to understand the biology of Aedes mosquitoes as it implies different inheritance history for nDNA and mtDNA or, as we have explained in the MS, some issue related to stochasticity. We could not test the latter because of the poor sampling of available genes. Obviously, one can sequence the genome of few dozens of Aedes and outgroups, but frankly this goes beyond the scope of our studies (see below). It is not a speculation that currently available multi-marker phylogenies and divergence times of mosquitoes should be taken with care until the discrepancy we have highlighted is completely understood.
Specific comments:
73 and 439-440
It is too general a statement. There are many parts of nuclear DNA, which are known to mutate faster than mtDNA e.g. microsatellites. Some introns or spacers show a similar rate of mutation.
DNA is not evolving – taxa evolve.
>Thanks for the observation. We corrected the sentence.
>As for “DNA is not evolving – taxa evolve” Well, it is an interesting statement. Based on our understanding of molecular evolution, we prefer to say that both evolves. Taxa are the (demographic) units on which selection act trough the environment. But it is the DNA variants that are selected; therefore a given allele has his own evolutionary history which in some case coincides with the taxa one, sometimes does not (because of incomplete linage sorting and complex gene family expansions).
113-114 I wonder if these markers (genes) were available from the same samples (specimens) for all or most of genes? If not - how Authors deal with this problem? Why Authors had not attempted to sequence missing genes for all or majority of Aedes species?
>We use genes from the same specimens whenever it was possible. In some cases we concatenate genes from different specimens, but obviously within exactly the same species. This is not a problem for inter-specific phylogenetic studies because the gene sorting happened at the speciation level. Anyway, this is the current state of dealing with the concatenation of markers: it has actually been done in the various literatures that we cite in the article. We have added a line of text to further explain this.
>We didn’t sequence missing genes because retrieving Aedes samples is a very complicated task: vast majority of them are from tropical countries and their collection requires enormous logistic effort. We would like to point that our study is a classic example of a detailed re-analysis of available data; it is not a study of sampling & sequencing. Often, a proper reanalysis of data reveal unexpected results which may have under looked in studies where most of the effort has been put in sampling. Indeed, this is the case of our MS. We are planning and granting a world scale sampling of Aedes for genome sequencing, but this requires a large effort involving partnering among various labs wolrdwide.
125-126 How many samples (specimens) were available for each of examined species? Based on the look on trees I suppose that only 1 / species. I think that it is not enough for many reasons. Especially if particular sequences were from various samples.
>In some cases all the genes came from one specimen, as in Aedes albopictus and Aedes aegypti, but for many specimens one gene correspond to a specimen. Again, this is the common approach in phylogenetic studies (please see the inter-specific phylogenetic studies cited in the MS which uses only one sequence for species). Our study, like the previous ones from other researchers, is an inter-specific phylogenetic study, not intra-specific. Therefore more individuals from the same species are not required.
128 How that ambiguous samples were determined?
>For each gene we run a fast phylogenetic analysis, when a sequence fall well outsides of their assigned group or known position we preferred to remove that sample.
325-327 I am surprised that divergence of taxa within the same genus could be so ancient. It is a common phenomenon in mosquitoes or Dipterans? Such ancient dates are determined by calibrations in very distant splits (e.g. flies and mosquitoes) – I am not convinced that such old dates are appropriate for calibration in studies within the single genus.
>In insect it is quite common to have ancient genus, particularly in genus with many generations per year such as Drosophila and mosquitoes. This is because of the (seemingly) high phenotypic similarity and the subsequent assignment of distantly related (genetically speaking) species to the same genus. This is actually one likely reason why mosquitoes have undergone a revision in their taxonomy.
>As for the calibrations, available fossils do not allow many close-to-present calibrations. We have extensively search the literature and indeed found minima to be used within the mosquitoes. The only safe maximum we could use is for the fly-mosquito split. We reassure reviewer that our approach is safe because in general we could obtain posterior estimates which are in line with previous ones (see our table 2).
I suppose that these estimates are not so reliable, especially as mutation rare in mosquitoes had to be c. one order of magnitude smaller than in other insects (see l. 441-442)
> Mutation rates (or more properly substitution rates) are known to change in different parts of a tree: this is because the rate varies at different time scale. Typically, mutation rate are higher close to tips because there have been fewer or no saturations. Rates become smaller the more we go into the past. The mitochondrial rate we infer for our dataset (Aedes rooted with flies) is typical of deep time scale: it is almost identical to the average mitochondrial rate inferred for the Ecdysozoan (the clade including arthropods and nematodes, references in the MS) something which reassure us that our rates and estimates are indeed very reliable. Another factor influencing the rate is the number of generation per year: this is why we take this as an indication that Aedes had a long history of fewer generations per year compared to Drosophila
Finally, I do not understand information about the divergence of taxa which are not closely related like species: A. aegypti and A. albopictus – these two species had no common recent ancestor, therefore the presentation of split dates for them is strange.
>A. aegypti and A. albopictus are both within the subgenus Stegomya and their split coincide with the split of the Stegomya clade, which is worth referencing. More importantly, these two species are by far the two most dangerous and most studied Aedes species, and the only two for which a genome has been sequenced. The age of their split is therefore important for comparative genomics (for example to calculate the rate of evolution of gene families)
395 I think that the whole paper should not be prepared without solving this problem. Currently, all possible explanations are speculative due to uncertainty in correct taxa assignment to sequences.
>As explained above, all taxa have been rigorously assigned according to their Genebank entry and following the normal concatenating procedure typically used in ours and other phylogenetic studies including those on Aedes (see for example reference 21 (Soghigian et al.) in the manuscript). We would like to stress that, in order to account for possible assignment issues, we repeated the clock analyses after excluding the few taxa that looked suspicious (see our Suppl. Figures S5-6): we obtain very similar results, therefore we conclude that incorrect assignment is not biasing our results, and therefore our interpretation of them.
407 I am surprised by so large missing data in the analyzed alignments – this could strongly affect results.
>We agree with the reviewer that missing data can affect the results. For this reason we have repeated all the analyses using another dataset (derived from reference 21) that is characterised by a much smaller amount of missing data (this is because it is centred on Aedes and does not have outgroups). Results are very consistent with those using our dataset indicating that “…the chronological discrepancy is not … affected by the amount of missing data in the datasets.” We have modified the text to make this clearer.
423-426, 481-483, 4870488, 492-494 I am surprised that Authors inform so many times that this study should be based on analyses of whole mitogenome or genome scans, what is true. So if they are aware of that need, why this study was not planned and executed on such data instead of the use of uncertain and leaky sequence datasets from previous studies?
>We performed various accessory analyses to exclude the possibility that our results were affected by 1) taxon sampling (fig S5-6), 2) missing data (fig S7), 3) saturation (fig S8). The only thing we could not test is the reduced number of genes. As we explained above (and to referee 2) we used the largest as possible dataset. The need to enlarge the gene sampling becomes therefore one of the main message of our manuscript. We would like to stress that the sampling, the sequencing, the genome/transcriptome assembly of circa 30 species is an extremely resource and time-consuming task. It will take years before we could organise a collection of Aedes and other mosquito’s specimens and perform proper genome sequencing: obviously this do not fit in our current manuscript.
In several places are genus and species names not italicized.
>Thank for this, we fixed that.
Reviewer 2 Report
As I noted in my comments in the pdf, I would like to see greater justification for using such a small number of genes. I understand the reasoning behind using only 4 mt genes if the remaining ~18 are not of sufficient quality across all taxa, but surely there are more than 4 high quality nuclear gene sequences available. Until this issue is resolved I find any conclusions/discussion of the detected incongruence suspect because it seems likely that the incongruence reflects sampling issues rather than biological/phylogenetic reality.

Author Response
As I noted in my comments in the pdf, I would like to see greater justification for using such a small number of genes. I understand the reasoning behind using only 4 mt genes if the remaining ~18 are not of sufficient quality across all taxa, but surely there are more than 4 high quality nuclear gene sequences available. Until this issue is resolved I find any conclusions/discussion of the detected incongruence suspect because it seems likely that the incongruence reflects sampling issues rather than biological/phylogenetic reality.
>Thanks for the many comments
>We used 4 nuclear genes because this was the best we could get from available annotated genes in database. The available nuclear genes (but also the mitochondrial ones) are really few for most of the taxa, in general no more than 5-6 types of nuclear markers per species, many species being represented only by one marker, usually COI. As we explained in the methods, in order to obtain a well-represented matrix we had to further exclude 3 candidate genes which were poorly represented in the outgroup to Aedes. Moreover, some available and suitable genes such as ITS were too fast evolving to be aligned below genus level. The only exceptions were Aedes albopictus and Aedes aegypti which have a genome sequenced. At the time of when we assembled the dataset there were some SRR of other mosquitoes but were mostly amplicon sequencing metagenomes (eg: SRX2175220, SRX2175221) not useful for marker extraction.
>About the gene sampling issue: we tried our best to disentagle the origin of the nucelar-mito incogruence. We performed various accessory analyses to exclude the possibility that our results were affected by 1) taxon sampling (fig S5-6), 2) missing data (fig S7), 3) saturation (fig S8). The only thing we could not test is the reduced number of genes and some type of stochastic error. As we explained above we used the larges as possible dataset given the current gene available. Therefore, the need to enlarge the gene sampling becomes one main messages of our manuscript. We are currently granting to organise a world scale sampling of Aedes for genome sequencing, but this requires a large effort involving partnering among various labs and years of research.
Does "more resolved" always equal "closer to reality?" Or does the ability to resolve a tree perhaps require outside evidence of the trees validity before assuming that the "more resolved" tree is "more correct?"
>We used “more resolved” sensu “with overall higher supports at nodes”. We agree with reviewer that more resolved does not equal to closer to reality. Indeed there are a lot of scientific evidences that proved that highly supported nodes are incorrect because they are the effect of undetected systematic errors (see reference 26 in the MS). We added a sentence to clarify this.
Where is fig 3a? I only see fig 3 which does not include the listed data.
>Corrected. Thanks, that was a typo, left because a previous version of the MS contained more panels in figure 3.
my confidence in these conclusions is colored by my suspicion that a data set larger than 8 genes would yield better results
>We totally agree with reviewer. We believe that adding a few extra genes (those that we have excluded because poorly sampled along the phylogeny) would not have yielded more signals, but rather more noise due to more gaps in the alignment. As we have explained above (and below), the main conclusion from our study is that the current gene sampling does not allow obtaining a clear picture of Aedes evolution. This is not a trivial statement because the community would think otherwise that an 8-genes well-supported phylogeny (as in our Figure 3) is enough for describing with confidence the Aedes evolution.
If you concluded that stochasticity was the true explanation, would you still want to publish the conclusions presented here? Perhaps there are additional statistical tests that would let you estimate the effect of stochasticity? or perhaps you could exploit fully the nuclear data available on ncbi, etc. to generate a tree based on a larger number of nuclear genes?
>We believe that our contribution is important even if we could not test genome-scaled data. We could exclude various possible misleading factors such as missing data and saturation (see our suppl. Figures S5-7); the only reaming possible factor that has to be tested is the amount of data. But as discussed above, there is not enough data in databases to build a larger, genome-scaled, dataset of Aedes. Only two Aedes have the genome annotated. There are some SSR samples from other Aedes, but they are mostly either amplicon data (ITS) or sRNA data. The other widely studied mosquito genus Anopheles is instead characterised by a large number of well curated genomes.
>Collecting specimens and generate genome-scaled data for a large panel of Aedes is a great effort that we believe goes beyond the scope of the current MS. Our scope was to point the community toward the previously undetected mito-nuclear incongruence, to provide as many as possible critical assessment of this incongruence, and to conclude that the current data is not enough for a proper evaluation of Aedes evolution. A part from Aedes, our study is important for the non Aedes community because it show that it is important to test the congruence of the phylogenetic signal in multi-marker studies. Too often, researcher concentrated on a concatenated dataset, without testing if the signal is similar in the two cytoplasmic sources of DNA.
Reviewer 3 Report
Congratulations, nice and interesting results. Some minor suggestions addressed directly in the text. Please check and refer the current data about fossils record of Culicidae and its interpretation - this will probably explain some of observations. Your molecular clock is not calibrated with fossils - it will be interesting to do it with larger material and also inclusion of sister groups as references.

Author Response
Congratulations, nice and interesting results. Some minor suggestions addressed directly in the text. Please check and refer the current data about fossils record of Culicidae and its interpretation - this will probably explain some of the observations. Your molecular clock is not calibrated with fossils - it will be interesting to do it with larger material and also the inclusion of sister groups as references.
>Thanks for appreciating our work.
>We have carefully checked the literature and the imprecision in numbering mosquitoes (these issues where presented in the pdf prepared by the referee) and modified accordingly.
>We further corrected to italics in two cases.
Be careful here with interpretation of molecular clock. This is true only if the assumption of immediate and parallel diversification of Culicomorpha in the early Jurassic is taken (but not supported with fossil data). Take a look on fossil record of Culicidae sister group - Chaoboridae - these are known since Toarcian, but oldest Culicidae much later, since Cenomanian.
>Thanks for these precious information, which we have reported in the manuscript.
Check some new references, it is large progress in this matter. The oldest unambiguous angiosperm fossils are pollen grains (~131.8 mya) from the the Early Cretaceous (Soltis et al. 2018), maybe there are some more recent findings, supporting this review: https://science.sciencemag.org/
>We add it to the reference, thanks
These dates are not calibrated with existing fossil record? It will be great to test these scenarios with fossil data first for better calibration of splits and with ecological requirements of taxa.
>All trees have been calibrated at three nodes with minima based on mosquitoes known fossils, plus we calibrated the root (see methods and references [41,42]). It is indeed our goal in future to build a more comprehensive dated phylogeny of Culicidae, but using more data in order to overcome stochastic types of errors.
This discrepancy could be indirect result of some major biotic changes in the Cenozoic (this could be reason for hybridization), but as you wrote without whole genome analysis at large scale it is difficult to assess.
>Yes, we agree that some major ecological transitions may have affected the mosquito radiation. However, as reviewer pointed out, until we have solved the puzzling chronological incongruence it is premature to propose paleo-ecological scenarios.
Round 2
Reviewer 1 Report
-
Reviewer 2 Report
Thank you for your thorough response to my comments.